# Group Equivariant Capsule Networks

**Jan Eric Lenssen**            **Matthias Fey**            **Pascal Libuschewski**

TU Dortmund University - Computer Graphics Group
44227 Dortmund, Germany
{janeric.lenssen, matthias.fey, pascal.libuschewski}@udo.edu

## Abstract

We present *group equivariant capsule networks*, a framework to introduce guaranteed equivariance and invariance properties to the capsule network idea. Our work can be divided into two contributions. First, we present a generic routing by agreement algorithm defined on elements of a group and prove that equivariance of output pose vectors, as well as invariance of output activations, hold under certain conditions. Second, we connect the resulting equivariant capsule networks with work from the field of group convolutional networks. Through this connection, we provide intuitions of how both methods relate and are able to combine the strengths of both approaches in one deep neural network architecture. The resulting framework allows sparse evaluation of the group convolution operator, provides control over specific equivariance and invariance properties, and can use routing by agreement instead of pooling operations. In addition, it is able to provide interpretable and equivariant representation vectors as output capsules, which disentangle evidence of object existence from its pose.

## 1 Introduction

Convolutional neural networks heavily rely on equivariance of the convolution operator under translation. Weights are shared between different spatial positions, which reduces the number of parameters and pairs well with the often occurring underlying translational transformations in image data. It naturally follows that a large amount of research is done to exploit other underlying transformations and symmetries and provide deep neural network models with equivariance or invariance under those transformations (*cf.* Figure 1). Further, equivariance and invariance are useful properties when aiming to produce data representations that disentangle factors of variation: when transforming a given input example by varying one factor, we usually aim for equivariance in one representation entry and invariance in the others. One recent line of methods that aim to provide a relaxed version of such a setting are *capsule networks*.

Our work focuses on obtaining a formalized version of capsule networks that guarantees those properties and bringing them together with *group equivariant convolutions* by Cohen and Welling [2016], which also provide provable equivariance properties under transformations within a group. In the following, we will shortly introduce capsule networks, as proposed by Hinton et al. and Sabour et al., before we outline our contribution in detail.

### 1.1 Capsule networks

Capsule networks [Hinton et al., 2011] and the recently proposed routing by agreement algorithm [Sabour et al., 2017] represent a different paradigm for deep neural networks for vision tasks. They aim to hard-wire the ability to disentangle the pose of an object from the evidence of its existence, also called *viewpoint equi- and invariance* in the context of vision tasks. This is done by encoding the output of one layer as a tuple of a pose vector and an activation. Further, they are

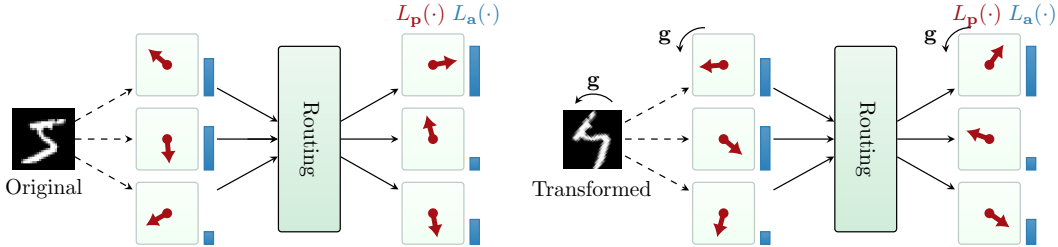

Equivariance of pose vectors: $L_\mathbf{p}(\mathbf{g} \circ \mathbf{P}, \mathbf{a}) = \mathbf{g} \circ L_\mathbf{p}(\mathbf{P}, \mathbf{a})$; Invariance of agreements: $L_\mathbf{a}(\mathbf{g} \circ \mathbf{P}, \mathbf{a}) = L_\mathbf{a}(\mathbf{P}, \mathbf{a})$

Figure 1: The task of dynamic routing for capsules with concepts of equivariant pose vectors and invariant agreements. Layers with those properties can be used to build viewpoint invariant architectures, which disentangle factors of variation.

inspired by human vision and detect linear, hierarchical relationships occurring in the data. Recent advances describe the dynamic routing by agreement method that iteratively computes how to route data from one layer to the next. One capsule layer receives $n$ pose matrices $\mathbf{M}_i$, which are then transformed by a trainable linear transformation $\mathbf{W}_{i,j}$ to cast $n$ votes for the pose of the $j$th output capsule:

$$\mathbf{V}_{i,j} = \mathbf{M}_i \cdot \mathbf{W}_{i,j}.$$

The votes are used to compute a proposal for an output pose by a variant of weighted averaging. The weights are then iteratively refined using distances between votes and the proposal. Last, an agreement value is computed as output activation, which encodes how strong the votes agree on the output pose. The capsule layer outputs a set of tuples $(\mathbf{M}, a)$, each containing the pose matrix and the agreement (as activation) of one output capsule.

## 1.2 Motivation and contribution

General capsule networks do not come with guaranteed equivariances or invariances which are essential to guarantee disentangled representations and viewpoint invariance. We identified two issues that prevent exact equivariance in current capsule architectures: First, the averaging of votes takes place in a vector space, while the underlying space of poses is a manifold. The vote averaging of vector space representations does not produce equivariant mean estimates on the manifold. Second, capsule layers use trainable transformation kernels defined over a local receptive field in the spatial vector field domain, where the receptive field coordinates are agnostic to the pose. They lead to non-equivariant votes and consequently, non-equivariant output poses. In this work, we propose possible solutions for these issues.

Our contribution can be divided into the following parts. First, we present group equivariant capsule layers, a specialized kind of capsule layer whose pose vectors are elements of a group $(G, \circ)$ (*cf.* Section 2). Given this restriction, we provide a general scheme for dynamic routing by agreement algorithms and show that, under certain conditions, equivariance and invariance properties under transformations from $G$ are mathematically guaranteed. Second, we tackle the issue of aggregating over local receptive fields in group capsule networks (*cf.* Section 3). Third, we bring together capsule networks with group convolutions and show how the group capsule layers can be leveraged to build convolutional neural networks that inherit the guaranteed equi- and invariances and produce disentangled representations (*cf.* Section 4). Last, we apply this combined architecture as proof of concept application of our framework to MNIST datasets and verify the properties experimentally.

## 2 Group equivariant capsules

We begin with essential definitions for group capsule layers and the properties we aim to guarantee. Given a Lie group $(G, \circ)$, we formally describe a group capsule layer with $m$ output capsules by a set of function tuples

$$\{(L_p^j(\mathbf{P}, \mathbf{a}), L_a^j(\mathbf{P}, \mathbf{a})) \mid j \in \{1, \dots, m\}\}. \tag{1}$$

Here, the functions $L_p$ compute the output pose vectors while functions $L_a$ compute output activations, given input pose vectors $\mathbf{P} = (\mathbf{p}_1, ..., \mathbf{p}_n) \in G^n$ and input activations $\mathbf{a} \in \mathbb{R}^n$. Since our goal

is to achieve global invariance and local equivariance under the group law $\circ$, we define those two properties for one single group capsule layer (*cf.* Figure 1). First, the function computing the output pose vectors of one layer is *left-equivariant* regarding applications of the group law if

$$L_p(\mathbf{g} \circ \mathbf{P}, \mathbf{a}) = \mathbf{g} \circ L_p(\mathbf{P}, \mathbf{a}), \quad \forall \mathbf{g} \in G. \tag{2}$$

Second, the function computing activations of one layer is *invariant* under applications of the group law $\circ$ if

$$L_a(\mathbf{g} \circ \mathbf{P}, \mathbf{a}) = L_a(\mathbf{P}, \mathbf{a}), \quad \forall \mathbf{g} \in G. \tag{3}$$

Since equivariance is transitive, it can be deducted that stacking layers that fulfill these properties preserves both properties for the combined operation. Therefore, if we apply a transformation from $G$ on the input of a sequence of those layers (*e.g.* a whole deep network), we do not change the resulting output activations but produce output pose vectors which are transformed by the same transformation. This sums up to fulfilling the vision of locally equivariant and globally invariant capsule networks.

## 2.1 Group capsule layer

We define the group capsule layer functions as the output of an iterative routing by agreement, similar to the approach proposed by Sabour et al. [2017]. The whole algorithm, given a generic *weighted average operation* $\mathcal{M}$ and a *distance measure* $\delta$, is shown in Algorithm 1.

---

**Algorithm 1** Group capsule layer

---

**Input**: poses $\mathbf{P} = (\mathbf{p}_1, \ldots, \mathbf{p}_n) \in G^n$, activations $\mathbf{a} = (a_1, \ldots, a_n) \in \mathbb{R}^n$
**Trainable parameters**: transformations $\mathbf{t}_{i,j}$
**Output**: poses $\hat{\mathbf{P}} = (\hat{\mathbf{p}}_1, \ldots, \hat{\mathbf{p}}_m) \in G^m$, activations $\hat{\mathbf{a}} = (\hat{a}_1, \ldots, \hat{a}_m) \in \mathbb{R}^m$

---

$\mathbf{v}_{i,j} \leftarrow \mathbf{p}_i \circ \mathbf{t}_{i,j}$          for all input capsules $i$ and output capsules $j$
$\hat{\mathbf{p}}_j \leftarrow \mathcal{M}((\mathbf{v}_{1,j}, \ldots, \mathbf{v}_{n,j}), \mathbf{a})$          $\forall j$
**for** $r$ iterations **do**
     $w_{i,j} \leftarrow \sigma(-\delta(\hat{\mathbf{p}}_j, \mathbf{v}_{i,j})) \cdot a_i$          $\forall i, j$
     $\hat{\mathbf{p}}_j \leftarrow \mathcal{M}((\mathbf{v}_{1,j}, \ldots, \mathbf{v}_{n,j}), \mathbf{w}_{:,j})$          $\forall j$
**end for**
$\hat{a}_j \leftarrow \sigma(-\frac{1}{n} \sum_{i=1}^{n} \delta(\hat{\mathbf{p}}_j, \mathbf{v}_{i,j}))$          $\forall j$
Return $\hat{\mathbf{p}}_1, \ldots, \hat{\mathbf{p}}_m, \hat{\mathbf{a}}$

---

Generally, votes are cast by applying trainable group elements $\mathbf{t}_{i,j}$ to the input pose vectors $\mathbf{p}_i$ (using the group law $\circ$), where $i$ and $j$ are the indices for input and output capsules, respectively. Then, the agreement is iteratively computed: First, new pose candidates are obtained by using the weighted average operator $\mathcal{M}$. Second, the negative, shifted $\delta$-distance between votes pose candidates are used for the weight update. Last, the agreement is computed by averaging negative distances between votes and the new pose. The functions $\sigma$ can be chosen to be some scaling and shifting non-linearity, for example $\sigma(x) = \texttt{sigmoid}(\alpha \cdot x + \beta)$ with trainable $\alpha$ and $\beta$, or as softmax over the output capsule dimension.

**Properties of $\mathcal{M}$ and $\delta$**    For the following theorems we need to define specific properties of $\mathcal{M}$ and $\delta$. The mean operation $\mathcal{M} : G^n \times \mathbb{R}^n \to G$ should map $n$ elements of the group $(G, \circ)$, weighted by values $\mathbf{x} = (x_1, ..., x_n) \in \mathbb{R}^n$, to some kind of weighted mean of those values in $G$. Besides the closure, $\mathcal{M}$ should be *left-equivariant* under the group law, formally:

$$\mathcal{M}(\mathbf{g} \circ \mathbf{P}, \mathbf{x}) = \mathbf{g} \circ \mathcal{M}(\mathbf{P}, \mathbf{x}), \quad \forall \mathbf{g} \in G, \tag{4}$$

and invariant under permutations of the inputs. Further, the distance measure $\delta$ needs to be chosen so that transformations $\mathbf{g} \in G$ are $\delta$-distance preserving:

$$\delta(\mathbf{g} \circ \mathbf{g}_1, \mathbf{g} \circ \mathbf{g}_2) = \delta(\mathbf{g}_1, \mathbf{g}_2), \mathbf{x}), \quad \forall \mathbf{g} \in G. \tag{5}$$

Given these preliminaries, we can formulate the following two theorems.

**Theorem 1.** *Let $\mathcal{M}$ be a weighted averaging operation that is equivariant under left-applications of $\mathbf{g} \in G$ and let $G$ be closed under applications of $\mathcal{M}$. Further, let $\delta$ be chosen so that all $\mathbf{g} \in G$ are $\delta$-distance preserving. Then, the function $L_p(\mathbf{P}, \mathbf{a}) = (\hat{\mathbf{p}}_1, \ldots, \hat{\mathbf{p}}_m)$, defined by Algorithm 1, is equivariant under left-applications of $\mathbf{g} \in G$ on input pose vectors $\mathbf{P} \in G^n$:*

$$L_p(\mathbf{g} \circ \mathbf{P}, \mathbf{a}) = \mathbf{g} \circ L_p(\mathbf{P}, \mathbf{a}), \qquad \forall \mathbf{g} \in G. \tag{6}$$

*Proof.* The theorem follows by induction over the inner loop of the algorithm, using the equivariance of $\mathcal{M}$, $\delta$-preservation and group properties. The full proof is provided in the appendix. □

**Theorem 2.** *Given the same conditions as in Theorem 1. Then, the function $L_a(\mathbf{P}, \mathbf{a}) = (\hat{a}_1, \ldots, \hat{a}_m)$ defined by Algorithm 1 is invariant under joint left-applications of $\mathbf{g} \in G$ on input pose vectors $\mathbf{P} \in G^n$:*

$$L_a(\mathbf{g} \circ \mathbf{P}, \mathbf{a}) = L_a(\mathbf{P}, \mathbf{a}), \quad \forall \mathbf{g} \in G. \tag{7}$$

*Proof.* The result follows by applying Theorem 1 and the $\delta$-distance preservation. The full proof is provided in the appendix. □

Given these two theorems (and the method proposed in Section 3), we are able to build a deep group capsule network, by a composition of those layers, that guarantees global invariance in output activations and equivariance in pose vectors.

## 2.2 Examples of useful groups

Given the proposed algorithm, $\mathcal{M}$ and $\delta$ have to be chosen based on the chosen group and element representations. A canonical application of the proposed framework on images is achieved by using the two-dimensional rotation group $SO(2)$. We chose to represent the elements of $G$ as two-dimensional unit vectors, $\mathcal{M}$ as the renormalized, Euclidean, weighted mean, and $\delta$ as the negative scalar product. Further higher dimensional groups include the three-dimensional rotation group $SO(3)$ and $GL(n, \mathbf{R})$, the group of general invertible matrices. Other potentially interesting applications of group capsules are translation groups. Further discussion about them, as well as the other groups, can be found in the appendix.

**Group products** It should be noted that using the direct product of groups allows us to apply our framework for group combinations. Given two groups $(G, \circ_G)$ and $(H, \circ_H)$, we can construct the direct product group $(G, \circ_G) \times (H, \circ_H) = (G \times H, \circ)$, with $(\mathbf{g}_1, \mathbf{h}_1) \circ (\mathbf{g}_2, \mathbf{h}_2) = (\mathbf{g}_1 \circ_G \mathbf{g}_2, \mathbf{h}_1 \circ_H \mathbf{h}_2)$. Thus, for example, the product $SO(2) \times (\mathbb{R}^2, +)$ is again a group. Therefore, Theorem 1 and 2 also apply for those combinations. As a result, the pose vectors contain independent poses for each group, keeping information disentangled between the individual ones.

## 3 Spatial aggregation with group capsules

This section describes our proposed spatial aggregation method for group capsule networks. As previously mentioned, current capsule networks perform spatial aggregation of capsules, which does not result in equivariant poses. When the input of a capsule network is transformed, not only the deeper pose vectors change accordingly. Since vector fields of poses are computed, the positions of those pose vectors in $\mathbb{R}^n$ might also change based on the transformation, formally modeled using the concept of induced representations [Cohen et al., 2018]. The trainable transformations $\mathbf{t}$ however, are defined for fixed positions of the local receptive field, which is agnostic to those translations. Therefore, the composition of pose vectors and trainable transformations to compute the votes depends on the input transformation, which prevents equivariance and invariance.

Formally, the votes $\mathbf{v}_i$ computed in a capsule layer over a local receptive field can be described by

$$\mathbf{v}_i = \mathbf{g} \circ p(\mathbf{g}^{-1}(\mathbf{x}_i)) \circ t(\mathbf{x}_i), \tag{8}$$

where $\mathbf{x}_i$ is a receptive field position, $p(\mathbf{x}_i)$ the input pose at position $\mathbf{x}_i$, $t(\mathbf{x}_i)$ the trainable transformation at position $\mathbf{x}_i$, and $\mathbf{g}$ the input transformation. It can be seen that we do not receive a set of equivariant votes $\mathbf{v}_i$ since the matching of $p(\cdot)$ and $t(\cdot)$ varies depending on $\mathbf{g}$. A visual example of the described issue (and a counterexample for equivariance) for an aggregation over a $2 \times 2$ block and $G = SO(2)$ can be found in Figures 2a and 2b.

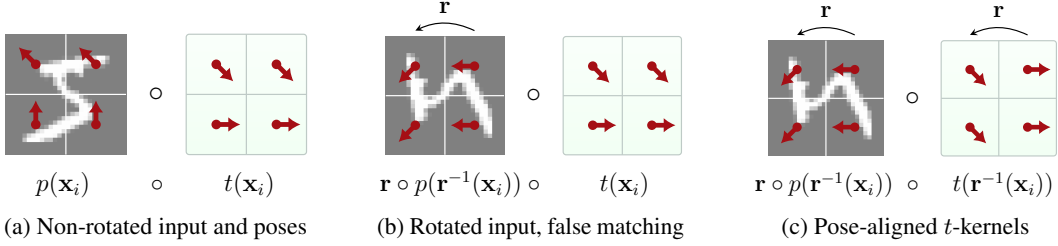

| $p(\mathbf{x}_i) \quad \circ \quad t(\mathbf{x}_i)$ | $\mathbf{r} \circ p(\mathbf{r}^{-1}(\mathbf{x}_i)) \circ \quad t(\mathbf{x}_i)$ | $\mathbf{r} \circ p(\mathbf{r}^{-1}(\mathbf{x}_i)) \quad \circ \quad t(\mathbf{r}^{-1}(\mathbf{x}_i))$ |
|:---:|:---:|:---:|
| (a) Non-rotated input and poses | (b) Rotated input, false matching | (c) Pose-aligned $t$-kernels |

Figure 2: Example for the spatial aggregation of a $2 \times 2$ block of $SO(2)$ capsules. Figure (a) shows the behavior for non-rotated inputs. The resulting votes have full agreement, pointing to the top. Figure (b) shows the behavior when rotating the input by $\pi/2$, where we obtain a different element-wise matching of pose vectors $p(\cdot)$ and transformations $t(\cdot)$, depending on the input rotation. Figure (c) shows the behavior with the proposed kernel alignment. It can be seen that $p$ and $t$ match again and the result is the same full pose agreement as in (a) with equivariant mean pose, pointing to the left.

**Pose-aligning transformation kernels** As a solution, we propose to align the constant positions $\mathbf{x}_i$ based on the pose before using them as input for a trainable transformation generator $t(\cdot)$. We can compute $\bar{\mathbf{p}} = \mathcal{M}(\mathbf{p}_1, \ldots, \mathbf{p}_n, \mathbf{1})$, a mean pose vector for the current receptive field, given local pose vectors $\mathbf{p}_1, \ldots, \mathbf{p}_n$. The mean poses of transformed and non-transformed inputs differ by the transformation $\mathbf{g}$: $\bar{\mathbf{p}} = \mathbf{g} \circ \bar{\mathbf{q}}$. This follows from equivariance of $\mathcal{M}$, invariance of $\mathcal{M}$ under permutation, and from the equivariance property of previous layers, meaning that the rotation applied to the input directly translates to the pose vectors in deeper layers. Therefore, we can apply the inverse mean pose $\bar{\mathbf{p}}^{-1} = \bar{\mathbf{q}}^{-1} \circ \mathbf{g}^{-1}$ to the constant input positions $\mathbf{x}$ of $t$ and calculate the votes as

$$\mathbf{v}_i = \mathbf{g} \circ p(\mathbf{g}^{-1}(\mathbf{x}_i)) \circ t((\bar{\mathbf{q}}^{-1} \circ \mathbf{g}^{-1})(\mathbf{x}_i)) = \mathbf{g} \circ p(\hat{\mathbf{x}}_i) \circ t(\bar{\mathbf{q}}^{-1}(\hat{\mathbf{x}}_i)), \qquad (9)$$

as shown as an example in Figure 2c. Using this construction, we use the induced representation as inputs for $p(\cdot)$ and $t(\cdot)$ equally, leading to a combination of $p(\cdot)$ and $t(\cdot)$ that is independent from $\mathbf{g}$. Note that $\bar{\mathbf{q}}^{-1} \in G$ is constant for all input transformations and therefore does not lead to further issues. In practice, we use a two-layer MLP to calculate $t(\cdot)$, which maps the normalized position to $n \cdot m$ transformations (for $n$ input capsules per position and $m$ output capsules). The proposed method can also be understood as pose-aligning a trainable, continuous kernel window, which generates transformations from $G$. It is similar to techniques applied for sparse data aggregation in irregular domains [Gilmer et al., 2017]. Since commutativity is not required, it also works for non-abelian groups (*e.g.* $SO(3)$). As an additional benefit, we observed significantly faster convergence during training when using the MLP generator instead of directly optimizing the transformations $\mathbf{t}$.

## 4 Group capsules and group convolutions

The newly won properties of pose vectors and activations allow us to combine our group equivariant capsule networks with methods from the field of group equivariant convolutional networks. We show that we can build sparse group convolutional networks that inherit invariance of activations under the group law from the capsule part of the network. Instead of using a regular discretization of the group, those networks evaluate the convolution for a fixed set of arbitrary group elements. The proposed method leads to improved theoretical efficiency for group convolutions, improves the qualitative performance of our capsule networks and is still able to provide disentangled information. In the following, we shortly introduce group convolutions before presenting the combined architecture.

**Group convolution** Group convolutions (G-convs) are a generalized convolution/correlation operator defined for elements of a group $(G, \circ)$ (here for Lie groups with underlying manifold):

$$[f \star \psi](\mathbf{g}) = \int_{\mathbf{h} \in G} \sum_{k=1}^{K} f_k(\mathbf{h}) \psi(\mathbf{g}^{-1}\mathbf{h}) \, d\mathbf{h}, \qquad (10)$$

for $K$ input feature signals, which behaves equivariant under applications of the group law $\circ$ [Cohen and Welling, 2016, Cohen et al., 2018]. The authors showed that they can be used to build group equivariant convolutional neural networks that apply a stack of those layers to obtain an equivariant architecture. However, compared to capsule networks, they do not directly compute disentangled representations, which we aim to achieve through the combination with capsule networks.

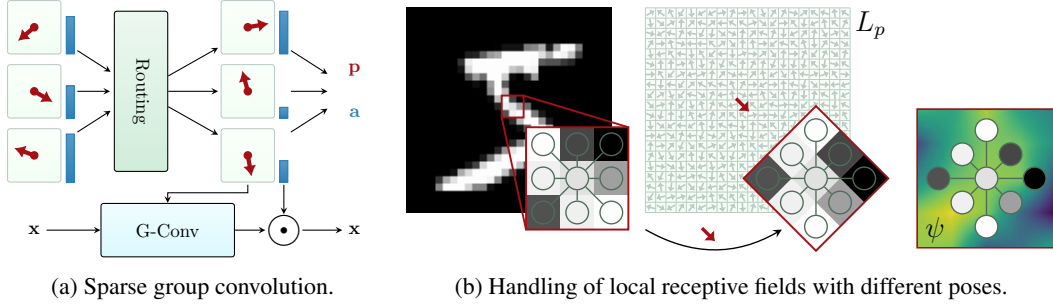

<div align="center">
(a) Sparse group convolution.      (b) Handling of local receptive fields with different poses.
</div>

Figure 3: (a) Scheme for the combination of capsules and group convolutions. Poses computed by dynamic routing are used to evaluate group convolutions. The output is weighted by the computed agreement. The invariance property of capsule activations is inherited to the output feature maps of the group convolutions. (b) Realization of the sparse group convolution. The local receptive fields are transformed using the calculated poses $L_p$ before aggregated using a continuous kernel function $\psi$.

## 4.1 Sparse group convolution

An intuition for the proposed method is to interpret our group capsule network as a sparse tree representation of a group equivariant network. The output feature map of a group convolution layer $[f \star \psi](\mathbf{g})$ over group $G$ is defined for each element $\mathbf{g} \in G$. In contrast, the output of our group capsule layer is a set of tuples $(\mathbf{g}, a)$ with group element $\mathbf{g}$ (pose vector) and activation $a$, which can be interpreted as a sparse index/value representation of the output of a G-conv layer. In this context, the pose $\mathbf{g}$, computed using routing by agreement from poses of layer $l$, serves as the hypothesis for the relevance of the feature map content of layer $l+1$ at position $\mathbf{g}$. We can now sparsely evaluate the feature map output of the group convolution and can use the agreement values from capsules to dampen or amplify the resulting feature map contents, bringing captured pose covariances into consideration. Figure 3a shows a scheme of this idea.

We show that when using the pose vector outputs to evaluate a G-conv layer for group element $\mathbf{g}$ we inherit the invariance property from the capsule activations, by proving the following theorem:

**Theorem 3.** *Given pose vector outputs $L_p(\mathbf{p}, \mathbf{a})$ of a group capsule layer for group $G$, input signal $f : G \to \mathbb{R}$, and filter $\psi : G \to \mathbb{R}$. Then, the group convolution $[f \star \psi]$ is invariant under joint left-applications of $\mathbf{g} \in G$ on capsule input pose vectors $\mathbf{P} \in G^n$ and signal $f$:*

$$[(\mathbf{g} \circ f) \star \psi] \left( L_p(\mathbf{g} \circ \mathbf{P}, \mathbf{a}) \right) = [f \star \psi] \left( L_p(\mathbf{P}, \mathbf{a}) \right). \tag{11}$$

*Proof.* The invariance follows from Theorem 1, the definition of group law application on the feature map, and the group properties. The full proof is provided in the appendix. □

The result tells us that when we pair each capsule in the network with an operator that performs pose-normalized convolution on a feature map, we get activations that are invariant under transformations from $G$. We can go one step further: given a group convolution layer for a product group, we can use the capsule output poses as an index for one group and densely evaluate the convolution for the other, leading to equivariance in the dense dimension (follows from equivariance of group convolution) and invariance in the capsule-indexed dimension. This leads to our proof of concept application with two-dimensional rotation and translation. We provide further formal details and a proof in the appendix.

Calculation of the convolutions can be performed by applying the inverse transformation to the local input using the capsule's pose vector, as it is shown in Figure 3b. In practice, it can be achieved, *e.g.*, by using the grid warping approach proposed by Henriques and Vedaldi [2017] or by using spatial graph-based convolution operators, *e.g.* from Fey et al. [2018]. Further, we can use the iteratively computed weights from the routing algorithm to perform *pooling by agreement* on the feature maps: instead of using max or average operators for spatial aggregation, the feature map content can be dynamically aggregated by weighting it with the routing weights before combining it.

# 5   Related work

Different ways to provide deep neural networks with specific equivariance properties have been introduced. One way is to share weights over differently rotated filters or augment the input heavily by transformations [Yanzhao et al., 2017, Weiler et al., 2018]. A related but more general set of methods are the group convolutional networks [Cohen and Welling, 2016, Dieleman et al., 2016] and its applications like Spherical CNNs in $SO(3)$ [Cohen et al., 2018] and Steerable CNNs in $SO(2)$ [Cohen and Welling, 2017], which both result in special convolution realizations.

Capsule networks were introduced by Hinton et al. [2011]. Lately, dynamic routing algorithms for capsule networks have been proposed [Sabour et al., 2017, Hinton et al., 2018]. Our work builds upon their methods and vision for capsule networks and connects those to the group equivariant networks.

Further methods include harmonic networks [Worrall et al., 2017], which use circular harmonics as a basis for filter sets, and vector field networks [Marcos et al., 2017]. These methods focus on two-dimensional rotational equivariance. While we chose an experiment which is similar to their approaches, our work aims to build a more general framework for different groups and disentangled representations.

# 6   Experiments

We provide proof of concept experiments to verify and visualize the theoretic properties shown in the previous sections. As an instance of our framework, we chose an architecture for rotational equivariant classification on different MNIST datasets [LeCun et al., 1998].

## 6.1   Implementation and training details

**Initial pose extraction**   An important subject which we did not tackle yet is the first pose extraction of a group capsule network. We need to extract pose vectors $\mathbf{p} \in G$ with activations $\mathbf{a}$ out of the raw input of the network without eliminating the equi- and invariance properties of Equations 2 and 3. Our solution for images is to simply compute local gradients using a Sobel operator and taking the length of the gradient as activation. For the case of a zero gradient, we need to ensure that capsules with only zero inputs also produce a zero agreement and an undefined pose vector.

**Convolution operator**   As convolution implementation we chose the spline-based convolution operator proposed by Fey et al. [2018]. Although the discrete two- or three-dimensional convolution operator is also applicable, this variant allows us to omit the resampling of grids after applying group transformations on the signal $f$. The reason for this is the continuous definition range of the B-spline kernel functions. Due to the representation of images as grid graphs, these kernels allow us to easily transform local neighborhoods by transforming the relative positions given on the edges.

**Dynamic routing**   In contrast to the method from Sabour et al. [2017], we do not use softmax over the output capsule dimension but the sigmoid function for each weight individually. The sigmoid function makes it possible for the network to route information to more than one output capsule and to no output capsule at all. Further, we use two iterations of computing pose proposals.

**Architecture and parameters**   Our canonical architecture consists of five capsule layers where each layer aggregates capsules from $2 \times 2$ spatial blocks with stride 2. The learned transformations are shared over the spatial positions. We use the routing procedure described in Section 2 and the spatial aggregation method described in Section 3. We also pair each capsule with a pose-indexed convolution as described in Section 4 with ReLU non-linearities after each layer, leading to a CNN architecture that is guided by pose vectors to become a sparse group CNN. The numbers of output capsules are 16, 32, 32, 64, and 10 per spatial position for each of the five capsule layers, respectively. In total, the architecture contains 235k trainable parameters (145k for the capsules and 90k for the CNN). The architecture results in two sets of classification outputs: the agreement values of the last capsule layer and the softmax outputs from the convolutional part. We use the spread loss as proposed by Hinton et al. [2018] for the capsule part and standard cross entropy loss for the convolutional part and add them up. We trained our models for 45 epochs. For further details, we refer to our implementation, which is available on Github[1].

|  | MNIST rot. (50k) | AffNist | MNIST rot. (10k) |  | Average pose error [degree] |
|---|---|---|---|---|---|
| CNN(*) | 92.30% | 81.64% | 90.19% | Naive average poses | 70.92 |
| Capsules | 94.68% | 71.86% | 91.87% | Capsules w/o recon. loss | 28.32 |
| **Whole** | **98.42%** | **89.10%** | **97.40%** | **Capsules with recon. loss** | **16.21** |

(a) Ablation experiment results           (b) Avg. pose errors for different configurations

Table 1: (a) Ablation experiments for the individual parts of our architecture including the CNN without induced pose vectors, the equivariant capsule network and the combined architecture. All MNIST experiments are conducted using randomly rotated training and testing data. (b) Average pose extraction error for three scenarios: simple averaging of initial pose vectors as baseline, our capsule architecture without reconstruction loss, and the same model with reconstruction loss.

## 6.2   Results

**Equivariance properties and accuracy**   We confirm equivariance and invariance properties of our architecture by training our network on non-rotated MNIST images and test it on images, which are randomly rotated by multiples of $\pi/2$. We can confirm that we achieve exactly the same accuracies, as if we evaluate on the non-rotated test set, which is $99.02\%$. We also obtain the same output activations and equivariant pose vectors with occasional small numerical errors $< 0.0001$, which confirms equi- and invariance. This is true for capsule and convolutional outputs. When we consider arbitrary rotations for testing, the accuracy of a network trained on non-rotated images is $89.12\%$, which is a decent generalization result, compared to standard CNNs.

For fully randomly rotated training and test sets we performed an ablation study using three datasets. Those include standard MNIST dataset with 50k training examples and the dedicated MNIST-rot dataset with the 10k/50k train/test split [Larochelle et al., 2007]. In addition, we replicated the experiment of Sabour et al. [2017] on the affNIST dataset[2], a modification of MNIST where small, random affine transformations are applied to the images. We trained on padded and translated (not rotated) MNIST and tested on affNIST. All results are shown in Table 1a. We chose our CNN architecture without information from the capsule part as our baseline (*). Without the induced poses, the network is equivalent to a traditional CNN, similar to the grid experiment presented by Fey et al. [2018]. When trained on a non-rotated MNIST, it achieves $99.13\%$ test accuracy and generalizes weakly to a rotated test set with only $58.79\%$ test accuracy. For training on rotated data, results are summarized in the table.

The results show that combining capsules with convolutions significantly outperforms both parts alone. The pose vectors provided by the capsule network guide the CNN, which significantly boosts the CNN for rotation invariant classification. We do *not* reach the state-of-the-art of $99.29\%$ in rotated MNIST classification obtained by Weiler et al. [2018]. In the affNIST experiment we surpass the result of 79% from Sabour et al. [2017] with much less parameters (235k vs. 6.8M) by a large margin.

**Representations**   We provide a quantitative and a qualitative analysis of generated representations of our MNIST trained model in Table 1b and Figure 4, respectively. We measured the average pose error by rotating each MNIST test example by a random angle and calculated the distance between the predicted and expected poses. The results of our capsule networks with and without a reconstruction loss (*cf.* next paragraph) are compared to the naive approach of hierarchically averaging local pose vectors. The capsule poses are far more accurate, since they do not depend equally on all local poses but mostly on those which can be explained by the existence of the detected object. It should be noted that the pose extraction was not directly supervised—the networks were trained using discriminative class annotations (and reconstruction loss) only. Similar to Sabour et al. [2017], we observe that using an additional reconstruction loss improves the extracted representations. In Figure 4a we show output poses for eleven random test samples, each rotated in $\pi/4$ steps. It can be seen that equivariant output poses are produced in most cases. The bottom row shows an error case, where an ambiguous pattern creates false poses. We provide a more detailed analysis for different MNIST classes in the appendix. Figure 4b shows poses after the first (top) and the second (bottom) capsule layer.

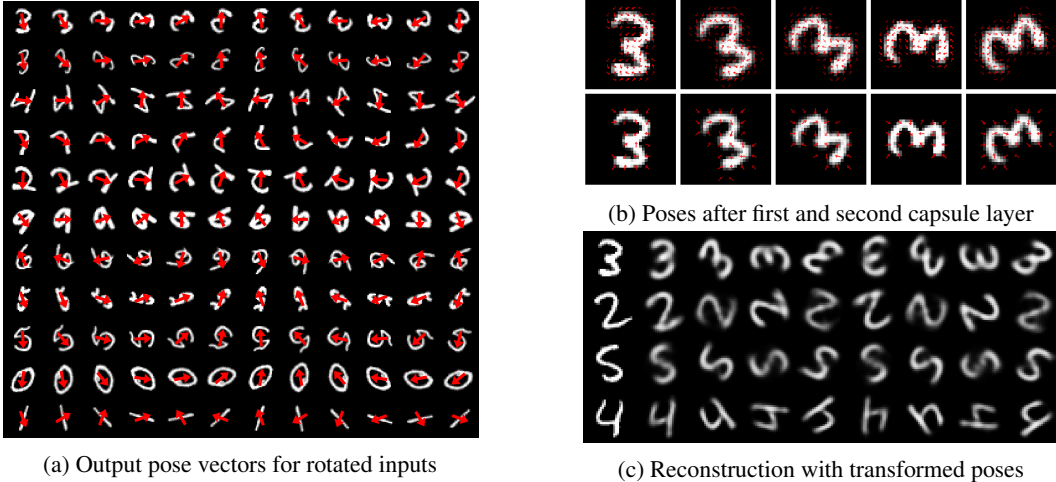

(a) Output pose vectors for rotated inputs

(b) Poses after first and second capsule layer

(c) Reconstruction with transformed poses

Figure 4: Visualization of output poses (a), internal poses (b), and reconstructions (c). (a) It can be seen that the network produces equivariant output pose vectors. The bottom row shows a rare error case, where symmetries lead to false poses. (b) Internal poses behave nearly equivariant, we can see differences due to changing discretization and image resampling. (c) The original test sample is on the left. Then, reconstructions after rotating the representation pose vector are shown. For the reconstruction, we selected visually correct reconstructed samples, which was not always the case.

**Reconstruction**   For further verification of disentanglement, we also replicated the autoencoder experiment of Sabour et al. [2017] by appending a three-layer MLP to convolution outputs, agreement outputs, and poses and train it to reconstruct the input image. Example reconstructions can be seen in Figure 4c. To verify the disentanglement of rotation, we provide reconstructions of the images after we applied $\pi/4$ rotations to the output pose vectors. It can be seen that we have fine-grained control over the orientation of the resulting image. However, not all representations were reconstructed correctly. We chose visually correct ones for display.

## 7   Limitations

Limitations of our method arise from the restriction of capsule poses to be elements of a group for which we have proper $\mathcal{M}$ and $\delta$. Therefore, in contrast to the original capsule networks, arbitrary pose vectors can no longer be extracted. Through product groups though, it is possible to combine several groups and achieve more general pose vectors with internally disentangled information if we can find $\mathcal{M}$ and $\delta$ for this group. For Lie groups, an implementation of an equivariant Karcher mean would be a sufficient operator for $\mathcal{M}$. It is defined as the point on the manifold that minimizes the sum of all weighted geodesic distances [Nielsen and Bhatia, 2012]. However, for each group there is a different number of possible realizations from which only few are applicable in a deep neural network architecture. Finding appropriate candidates and evaluating them is part of our future work.

## 8   Conclusion

We proposed group equivariant capsule networks that provide provable equivariance and invariance properties. They include a scheme for routing by agreement algorithms, a spatial aggregation method, and the ability to integrate group convolutions. We proved the relevant properties and confirmed them through proof of concept experiments while showing that our architecture provides disentangled pose vectors. In addition, we provided an example of how sparse group equivariant CNNs can be constructed using guiding poses. Future work will include applying the proposed framework to other, higher-dimensional groups, to come closer to the expressiveness of original capsule networks while preserving the guarantees.

**Acknowledgments**

Part of the work on this paper has been supported by Deutsche Forschungsgemeinschaft (DFG) within the Collaborative Research Center SFB 876 *Providing Information by Resource-Constrained Analysis*, projects B2 and A6.

## Footnotes

[1]Implementation at: `https://github.com/mrjel/group_equivariant_capsules_pytorch`

[2]affNIST: `http://www.cs.toronto.edu/~tijmen/affNIST/`

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
