[Supplementary Material]

# Appendix – Group Equivariant Capsule Networks

**Jan Eric Lenssen**　　　　**Matthias Fey**　　　　**Pascal Libuschewski**

In the following, we provide the detailed proofs for Theorems 1, 2, and 3 in Section 1, further information about applicable groups in Section 2, a formal presentation of capsule convolution with product groups in Section 3, and a quantitative analysis of pose vectors in Section 4.

## 1　Proofs for theorems

Because the theorems and proofs refer to Algorithm 1, we present it again:

---
**Algorithm 1** Group capsule layer

---
**Input**: poses $\mathbf{P} = (\mathbf{p}_1, \ldots, \mathbf{p}_n) \in G^n$, activations $\mathbf{a} = (a_1, \ldots, a_n) \in \mathbb{R}^n$
**Trainable parameters**: transformations $\mathbf{t}_{i,j}$
**Output**: poses $\hat{\mathbf{P}} = (\hat{\mathbf{p}}_1, \ldots, \hat{\mathbf{p}}_m) \in G^m$, activations $\hat{\mathbf{a}} = (\hat{a}_1, \ldots, \hat{a}_m) \in \mathbb{R}^m$

---
$\mathbf{v}_{i,j} \leftarrow \mathbf{p}_i \circ \mathbf{t}_{i,j}$　　　　　　　　　　for all input capsules $i$ and output capsules $j$
$\hat{\mathbf{p}}_j \leftarrow \mathcal{M}((\mathbf{v}_{1,j}, \ldots, \mathbf{v}_{n,j}), \mathbf{a})$　　　　　　　　　　　　　　　　　　　　$\forall j$
**for** $r$ iterations **do**
　$w_{i,j} \leftarrow \sigma(-\delta(\hat{\mathbf{p}}_j, \mathbf{v}_{i,j})) \cdot a_i$　　　　　　　　　　　　　　　　　$\forall i, j$
　$\hat{\mathbf{p}}_j \leftarrow \mathcal{M}((\mathbf{v}_{1,j}, \ldots, \mathbf{v}_{n,j}), \mathbf{w}_{:,j})$　　　　　　　　　　　　　　　　$\forall j$
**end for**
$\hat{a}_j \leftarrow \sigma(-\frac{1}{n} \sum_{i=1}^n \delta(\hat{\mathbf{p}}_j, \mathbf{v}_{i,j}))$　　　　　　　　　　　　　　　　　　$\forall j$
Return $\hat{\mathbf{p}}_1, \ldots, \hat{\mathbf{p}}_m, \hat{\mathbf{a}}$

---

**Theorem 1.** *Let $\mathcal{M}$ be a weighted averaging operation that is equivariant under left-applications of $\mathbf{g} \in G$ and let $G$ be closed under applications of $\mathcal{M}$. Further, let $\delta$ be chosen so that all $\mathbf{g} \in G$ are $\delta$-distance preserving. Then, the function $L_p(\mathbf{P}, \mathbf{a}) = (\hat{\mathbf{p}}_1, \ldots, \hat{\mathbf{p}}_m)$ defined by Algorithm 1 is equivariant under left-applications of all $\mathbf{g} \in G$ on input pose vectors $\mathbf{P} \in G^n$:*

$$L_p(\mathbf{g} \circ \mathbf{P}, \mathbf{a}) = \mathbf{g} \circ L_p(\mathbf{P}, \mathbf{a}), \qquad \forall \mathbf{g} \in G. \tag{1}$$

*Proof.* The theorem is shown by induction over the inner loop of the algorithm, using the equivariance of $\mathcal{M}$, preservation of $\delta$ and group properties. The initial step is to show equivariance of the pose vectors before the loop. After that we show that, given equivariant first pose vectors we receive invariant routing weights $\mathbf{w}$, which again leads to equivariant pose vectors in the next iteration.

*Induction Basis.* Let $\hat{\mathbf{p}}_0$, $\hat{\mathbf{p}}_0^{\mathbf{g}}$ be the first computed pose vectors (before the loop) for non-transformed and transformed inputs, respectively. The equivariance of those poses can be shown given associativity of group law, the equivariance of $\mathcal{M}$ and the invariance of activations coming from a previous layer (input activations $\mathbf{a}$ are equal for transformed and non transformed inputs). Note that we show the result for one output capsule. Therefore, index $j$ is constant and omitted.

$$
\begin{aligned}
\hat{\mathbf{p}}_0^{\mathbf{g}} &= \mathcal{M}(((\mathbf{g} \circ \mathbf{p}_1) \circ \mathbf{t}_1, \ldots, (\mathbf{g} \circ \mathbf{p}_n) \circ \mathbf{t}_n), \mathbf{a}^{\mathbf{g}}) \\
&= \mathcal{M}((\mathbf{g} \circ (\mathbf{p}_1 \circ \mathbf{t}_1)), \ldots, \mathbf{g} \circ (\mathbf{p}_n \circ \mathbf{t}_n)), \mathbf{a}) \\
&= \mathbf{g} \circ \mathcal{M}((\mathbf{p}_1 \circ \mathbf{t}_1, \ldots, \mathbf{p}_n \circ \mathbf{t}_n), \mathbf{a}) \\
&= \mathbf{g} \circ \hat{\mathbf{p}}_0
\end{aligned}
$$

In addition, it is clear to see that the computed votes also are equivariant.

*Induction Step.* Assuming equivariance of old pose vectors $(\mathbf{g} \circ \hat{\mathbf{p}}_m = \hat{\mathbf{p}}_m^{\mathbf{g}})$, we show equivariance of new pose vectors $(\mathbf{g} \circ \hat{\mathbf{p}}_{m+1} = \hat{\mathbf{p}}_{m+1}^{\mathbf{g}})$ after the next routing iteration. First we show that calculated weights $\mathbf{w}$ behave again invariant under input transformation $\mathbf{g}$. This follows directly from the induction assumption, $\delta$-distance preservation, equivariance of the votes and the invariance of $\mathbf{a}$:

$$
\begin{aligned}
w_i^{\mathbf{g}} &= \sigma(-\delta(\hat{\mathbf{p}}_m^{\mathbf{g}}, \mathbf{v}_i^{\mathbf{g}})) \cdot a_i \\
&= \sigma(-\delta(\mathbf{g} \circ \hat{\mathbf{p}}_m, \mathbf{g} \circ \mathbf{v}_i)) \cdot a_i \\
&= \sigma(-\delta(\hat{\mathbf{p}}_m, \mathbf{v}_i)) \cdot a_i \\
&= w_i
\end{aligned}
$$

Now we show equivariance of $\hat{\mathbf{p}}_{m+1}$, similarly to the induction basis, but using invariance of $\mathbf{w}$:

$$
\begin{aligned}
\hat{\mathbf{p}}_{m+1}^{\mathbf{g}} &= \mathcal{M}(((\mathbf{g} \circ \mathbf{p}_1) \circ \mathbf{t}_1, \ldots, (\mathbf{g} \circ \mathbf{p}_n) \circ \mathbf{t}_n), \mathbf{w}^{\mathbf{g}}) \\
&= \mathcal{M}((\mathbf{g} \circ (\mathbf{p}_1 \circ \mathbf{t}_1)), \ldots, \mathbf{g} \circ (\mathbf{p}_n \circ \mathbf{t}_n)), \mathbf{w}) \\
&= \mathbf{g} \circ \mathcal{M}((\mathbf{p}_1 \circ \mathbf{t}_1, \ldots, \mathbf{p}_n \circ \mathbf{t}_n), \mathbf{w}) \\
&= \mathbf{g} \circ \hat{\mathbf{p}}_{m+1}
\end{aligned}
$$

$\square$

**Theorem 2.** *Given the same conditions as in Theorem 1. Then, the function* $L_a(\mathbf{P}, \mathbf{a}) = (\hat{a}_1, \ldots, \hat{a}_m)$ *defined by Algorithm 1 is invariant under joint left-applications of* $\mathbf{g} \in G$ *on input pose vectors* $\mathbf{P} \in G^n$:

$$
L_a(\mathbf{g} \circ \mathbf{P}, \mathbf{a}) = L_a(\mathbf{P}, \mathbf{a}), \quad \forall \mathbf{g} \in G. \tag{2}
$$

*Proof.* The result follows by applying Theorem 1 and the $\delta$-distance preservation. Equality of $a$ and $a^{\mathbf{g}}$ is shown using Theorem 1 and the $\delta$-distance preservation of $G$. Again, wie show the result for one output capsule. The $\sigma$ is constant and therefore omitted for simplicity.

$$
\begin{aligned}
a^{\mathbf{g}} &= \sum_{i=1}^{n} \delta(L_p(\mathbf{g}_l \circ \mathbf{P}, \mathbf{a}), \mathbf{g}_l \circ \mathbf{p}_i \circ \mathbf{g}_i) \\
&= \sum_{i=1}^{n} \delta(\mathbf{g}_l \circ L_p(\mathbf{P}, \mathbf{a}), \mathbf{g}_l \circ \mathbf{p}_i \circ \mathbf{g}_i) \\
&= \sum_{i=1}^{n} \delta(L_p(\mathbf{P}, \mathbf{a}), \mathbf{p}_i \circ \mathbf{g}_i) \\
&= a
\end{aligned}
$$

$\square$

**Theorem 3.** *Given pose vector outputs* $L_p(\mathbf{p}, \mathbf{a})$ *of a group capsule layer for group* $G$, *input signal* $f : G \to \mathbb{R}$ *and filter* $\psi : G \to \mathbb{R}$. *Then, the group convolution* $[f \star \psi]$ *is invariant under joint left-applications of* $\mathbf{g} \in G$ *on capsule input pose vectors* $\mathbf{P} \in G^n$ *and signal* $f$:

$$
[(\mathbf{g} \circ f) \star \psi] (L_p(\mathbf{g} \circ \mathbf{P}, \mathbf{a})) = [f \star \psi] (L_p(\mathbf{p}, \mathbf{a})) \tag{3}
$$

*Proof.* The result is shown by applying Theorem 1, the definition of group law application on the feature map $(\mathbf{g} \circ f)(\mathbf{h}) = f(\mathbf{g}^{-1} \circ \mathbf{h})$, a substitution $\mathbf{h} \to \mathbf{g} \cdot \mathbf{h}$ and the group property $(\mathbf{g}_1 \circ \mathbf{g}_2)^{-1} =$

$\mathbf{g}_2^{-1} \circ \mathbf{g}_1^{-1}$ (using existence of inverse and neutral element properties of groups):

$$[(\mathbf{g} \circ f) \star \psi] (L_p(\mathbf{g} \circ \mathbf{P}, \mathbf{a})) = [(\mathbf{g} \circ f) \star \psi] (\mathbf{g} \circ L_p(\mathbf{P}, \mathbf{a}))$$

$$= \int_{\mathbf{h} \in G} \sum_i f_i(\mathbf{g}^{-1} \circ \mathbf{h}) \psi_i((\mathbf{g} \circ L_p(\mathbf{P}, \mathbf{a}))^{-1} \circ \mathbf{h}) \, d\mathbf{h}$$

$$= \int_{\mathbf{h} \in G} \sum_i f_i(\mathbf{h}) \psi_i((\mathbf{g} \circ L_p(\mathbf{P}, \mathbf{a}))^{-1} \circ \mathbf{g} \circ \mathbf{h}) \, d\mathbf{h}$$

$$= \int_{\mathbf{h} \in G} \sum_i f_i(\mathbf{h}) \psi_i((L_p(\mathbf{P}, \mathbf{a})^{-1} \circ \mathbf{g}^{-1} \circ \mathbf{g} \circ \mathbf{h}) \, d\mathbf{h}$$

$$= \int_{\mathbf{h} \in G} \sum_i f_i(\mathbf{h}) \psi_i((L_p(\mathbf{P}, \mathbf{a})^{-1} \circ \mathbf{h}) \, d\mathbf{h}$$

$$= [f \star \psi] (L_p(\mathbf{p}, \mathbf{a}))$$

$\square$

## 2 Examples for useful groups

Given the proposed algorithm, $\mathcal{M}$ and $\delta$ have to be chosen based on the chosen group and element representations. Here we provide more information about Lie groups which provide useful equivariances and can potentially be used in our framework.

**The two-dimensional rotation group** $SO(2)$   The canonical application of the proposed framework on images is achieved by using the two-dimensional rotation group $SO(2)$. We chose to represent the elements of $G$ as two-dimensional unit vectors, chose $\mathcal{M}$ as the renormalized Euclidean weighted mean and $\delta$ as the negative scalar product. Then, $\delta$ is distance preserving and $\mathcal{M}$ is left-equivariant, assuming given poses do not add up to zero, which can be guaranteed through practical measures.

**Translation group** $(\mathbb{R}^n, +)$   An potentially interesting application of group capsules are translation groups. Essentially, a layer in the network is no longer evaluated for each spatial position, but rather predict which positions will be of special interest and may sparsely evaluate a feature map at those points. Therefore, the number of evaluations is heavily reduced, from number of output capsules times number of pixels in the feature map to only the number of output capsules. However, in our current architectures we would not expect that this construction would work, because the capsule network would not be able to receive gradients which point in the direction of good transformations $\mathbf{t}$. It would rather be a random search, until good translational dependencies between hierarchical parts of objects are found. Also, due to usually local filters in convolutions and sparse evaluations, the outputs would often be zero at points of interest. Choosing $\mathcal{M}$ and $\delta$ however is straight-forward: the Euclidean weighted average and the $l2$-distance fulfill all requirements.

**Higher dimensional groups**   Further interesting groups include the three-dimensional rotation group $SO(3)$ as well as $GL(n, \mathbf{R})$, the group of general invertible matrices. For $SO(3)$, a sufficient averaging operator $\mathcal{M}$ would be the weighted, element-wise mean of rotation matrices, orthogonally projected onto the $SO(3)$, as was shown by Moakher [2002] (it is not trivial to compute this operator in a differentiable neural network module, though). Distance $\delta$ can be chosen as the Frobenius distance, as rotation matrices are (Euclidean-)distance preserving.

## 3 Product group convolutions

Using the direct product of groups allows to apply our framework for group combinations. For example, the product $SO(2) \times (\mathbb{R}^2, +)$ is again a group. Therefore, Theorem 1 and 2 also apply for those combinations. Acknowledging that, we can go further and only use capsule routing for a subset of groups in the product: Given a product group $(G, \circ) = (G_1, \circ_1) \times (G_2, \circ_2)$ (note that both can again be product groups), we can use routing by agreement with sparse convolution evaluation

over the group $(G_1, \circ_1)$ and dense convolution evaluation without routing over the group $(G_2, \circ_2)$. Evaluation of the convolution operator changes to

$$[f \star \psi](\hat{\mathbf{r}}, \mathbf{t}) = \int_{(\mathbf{g}_1, \mathbf{g}_2) \in G} \sum_i f_i(\mathbf{g}_1, \mathbf{g}_2) \psi_i(\hat{\mathbf{r}}^{-1} \circ \mathbf{g}_1, \mathbf{t}^{-1} \circ \mathbf{g}_2) \, d\mathbf{g}_1 d\mathbf{g}_2. \qquad (4)$$

We preserve equi- and invariance results for the group with routing and equivariance for the one without, which we show by proving the following theorem. For the given example $SO(2) \times (\mathbb{R}^2, +)$, it leads to evaluating the feature maps densely for spatial translations while sparsely evaluating different rotations at each position and routing between them from a layer $l$ to layer $l+1$. Activations would still be invariant under application of the group that is indexed by the capsule poses.

**Theorem 4.** *Let $(G, \circ) = (R, \circ_1) \times (T, \circ_2)$ be a direct product group and $\mathcal{M}$ and $\delta$ be given like in Theorem 1. Further, let $\mathbf{e}$ be the neutral element of group $T$. Then, the group convolution $[f \star \psi]$ is invariant under joint left-applications of $\mathbf{r} \in R$ on capsule input pose vectors $\mathbf{P} \in R^n$ and signal $f$, for all $\mathbf{t} \in T$:*

$$[(\mathbf{r}, \mathbf{e}) \circ f \star \psi] (L_p(\mathbf{r} \circ \mathbf{P}, \mathbf{a}), \mathbf{t}) = [f \star \psi] (L_p(\mathbf{P}, \mathbf{a}), \mathbf{t}) \qquad (5)$$

*Proof.* The proof is given for one output capsule $j$ and one input feature map $i$ (omitting the sum in the process). We show the equality analogously to Theorem 3 by applying the result of Theorem 1, the definition of group law application on the feature map $((\mathbf{g}_1, \mathbf{g}_2) \circ f)(\mathbf{h}_1, \mathbf{h}_2) = f(\mathbf{g}_1^{-1} \circ \mathbf{h}_1, \mathbf{g}_1^{-1} \circ \mathbf{h}_2)$, a substitution $\mathbf{g}_1 \to \mathbf{r} \cdot \mathbf{g}_1$ and the group property $(\mathbf{g}_1 \circ \mathbf{g}_2)^{-1} = \mathbf{g}_2^{-1} \circ \mathbf{g}_1^{-1}$, using the existence of inverse and neutral element properties of groups (omitting $d\mathbf{g}$'s):

$$[(\mathbf{r}, \mathbf{e}) \circ f \star \psi] (L_p(\mathbf{r} \circ \mathbf{P}, \mathbf{a}), \mathbf{t})$$
$$= [(\mathbf{r}, \mathbf{e}) \circ f \star \psi] (\mathbf{r} \circ L_p(\mathbf{P}, \mathbf{a}), \mathbf{t})$$
$$= \int_{(\mathbf{g}_1, \mathbf{g}_2) \in G} f(\mathbf{r}^{-1} \circ \mathbf{g}_1, \mathbf{e} \circ \mathbf{g}_2) \psi((\mathbf{r} \circ L_p(\mathbf{P}, \mathbf{a}))^{-1} \circ \mathbf{g}_1, \mathbf{t}^{-1} \circ \mathbf{g}_2)$$
$$= \int_{(\mathbf{g}_1, \mathbf{g}_2) \in G} f(\mathbf{g}_1, \mathbf{g}_2) \psi((\mathbf{r} \circ L_p(\mathbf{P}, \mathbf{a}))^{-1} \circ \mathbf{r} \circ \mathbf{g}_1, \mathbf{t}^{-1} \circ \mathbf{g}_2)$$
$$= \int_{(\mathbf{g}_1, \mathbf{g}_2) \in G} f(\mathbf{g}_1, \mathbf{g}_2) \psi((L_p(\mathbf{P}, \mathbf{a})_1^{-1} \circ \mathbf{r}^{-1} \circ \mathbf{r} \circ \mathbf{g}_1, \mathbf{t}^{-1} \circ \mathbf{g}_2)$$
$$= \int_{(\mathbf{g}_1, \mathbf{g}_2) \in G} f(\mathbf{g}_1, \mathbf{g}_2) \psi((L_p(\mathbf{P}, \mathbf{a})_1^{-1} \circ \mathbf{g}_1, \mathbf{t}^{-1} \circ \mathbf{g}_2)$$
$$= [f \star \psi] (L_p(\mathbf{P}, \mathbf{a}), \mathbf{t})$$

$\square$

The theorem allows us to create several types of capsule modules which precisely allow to choose equivariances and invariances over a set of groups. Looking again at the example $SO(2) \times (\mathbb{R}^2, +)$, Theorem 4 also shows us a convenient procedure: The convolution for each spatial position $\mathbf{t}$ and sparse rotation $\mathbf{r}$ can be computed by rotating a local window of the input feature map by $\mathbf{r}$ before applying the convolution. Therefore, we obtain a straight-forward way to implement CNNs with guaranteed rotational invariance (which also outputs pose vectors with guaranteed equivariance).

It should be noted that we do not use the roto-translation group $SE(2)$ here, which is a group over entangled rotation and translation. Though we expect that modeling with this group is also possible, the proofs and the concepts are simpler, when using a direct product. The reason for this is that we aim to use both parts in different ways and to keep information disentangled.

## 4   Quantitative analysis of pose vectors

In the main paper we showed that the exact equivariance of pose vectors can be confirmed when only considering rotations by multiples of $\pi/2$. However, when rotating by different angles, the image gets resampled. This leads to an error in pose vectors for arbitrary rotations, which was evaluated quantitatively in the paper.. We plotted this error for each MNIST class individually in Figure 1. It can be seen that, for all classes, far away predictions are rarer than those near the correct pose. We

Figure 1: Angle error histograms for rotated inputs that require resampling. The plots are given for each MNIST class individually. The $x$ axis shows bins for the angle errors in degree. The $y$ axis represents the fraction of test examples falling in each bin.

can also observe variances between the classes. The classes with the largest errors are 1, 4 and 8 while pose vectors from classes 3, 6 and 9 are most accurate. We suspect that inherent symmetries of the symbols cause a larger pose error.

## References

M. Moakher. Means and averaging in the group of rotations. *SIAM Journal on Matrix Analysis and Applications (SIMAX)*, 24(1):1–16, 2002.