[Reviews · NeurIPS 2018]

Reviewer 1



Authors present a modification of Capsule networks which guarantees equivarience to SO(2) group of transformations. Since restricting the pose matrices of a capsule network to operate inside the group degrades the performance of the network, they also suggest a method for combining group convolutional layers with capsule layers. Although the theoretical aspect of this work is strong, but experimental evaluations are quite limited without a proper comparison to baselines andc other works. Pros: The paper is well written and conveys the idea clearly. Capsule networks were proposed initially with the promise of better generalization in terms of affine transformations and viewpoint invarience. This work is a step toward guaranteed viewpoint invarience which explains the theoretical language and the concepts that the network should abide by. Cons: Significance of the paper is vague and under question due to limited experimental setup. The architecture that is used in this paper has many more parameters in compare to Sabour et al (5 capsule layers vs 1, plus additional group convolutional layers). I would suggest for the Affnist generalizability test to compare at least with a baseline of their own or have an ablation study. Specially because in Hinton 2018 (Matrix Capsules with EM routing) authors report a generalizability to affnist of 6.9% test error (in the rebuttal section) which is significantly better than 11% test error reported here. The gain from group convolution layers is not clear. The paper can benefit from a comparison with a comparable CapsNet and a comparison with group convolution network or steerable cnns as a baseline. The significant gap between testing on multiples of pi/2 vs random rotations is also concerning. The goal of rotation invarience is with respect to all random rotations and this limitation has to be addressed. More elaboration in this matter is necesssary. For the claim on 1 & 7, 6 & 9, the authors can compare the confusion matrices to verify that the poor results is actually due to significant increase of confusion between 6s & 9s. The performance drop on MNIST (from 99.75 to 99.02) even with the group convolution layers and significantly larger network is not justified. Furthermore, reporting the results on at least one more toy dataset is absolutely necessary, for example small norb. Also the added computation cost should be mentioned in the paper. The significance of this work would be much smaller due to the limitations from accuracy and computation costs. --------------------------------------------Update Thanks for the comparison with the baselines and providing the number of trainable parameters. The effectiveness of the method is more clear now and I'm increasing my score to 7 considering the addition of extra ablation study. Suggestions to further improve the exposition: - Extra ablation study on the choices made in the paper like using sigmoid rather than softmax for routing would improve the experimental setup and number of routing iterations. ShapeNet could also be a good candidate for extra experiments (limited to planar rotations, etc). - Clarifying the pose normalization write up by distinguishing the Capsule pose parameters from the spatial kernel positions. For example use different words to refer to this two concepts to reduce the confusion (e.g. Receptive Field Coordinate (RFC) vs pose rather than constant spatial position). - Expanding on how Capsule poses are used to index into GCN that would clarify the differences with the original GCN paper.

Reviewer 2



While previous work on capsules [1,2,3] identifies equivariance of the routing process as a desirable property, these methods do not guarantee equivariance. The goal if this paper is to create capsule networks with built-in equivariance. The paper contains several interesting ideas, but lacks sufficient theoretical motivation and experimental support for the proposed architectural innovations. Several more or less independent changes are proposed (equivariant routing, pose normalization layers, spatial aggregation, combination with group convolution), and although each addition is motivated by a plausible sounding line of reasoning, ultimately the proposed innovations don't seem innevitable, and no rigorous ablation study is performed. Although section 2 is very clearly written, I found it hard to understand section 3. It is not clear to me why simply summing the vectors in Fig 2b would be a problem (the result would be equivariant). Figure 2c seems like a very unnatural way to represent the data. It seems like a lot of complexity in this paper can be removed by working with representations of G instead of G itself. Specifically, if the pose vectors (which is a misnomer if they are group elements) are taken to be vectors in a vector space V in which a (unitary) representation of G is defined, then M and delta can be simply defined as the vector mean and inner product, and the required properties will follow automatically (and this works for any group, not just SO(2)). The issue identified in section 3, where each vector is seen to rotate and move to a new place, can then be understood very simply as in [4] via the notion of induced representation. I find it strange that the authors write "Since we take expressiveness from the trainable pose transformations to ensure equivariance and invariance, it is not to be expected that the proposed capsule networks using the proposed routing algorithms will achieve the same classification accuracies as their original counterparts" If these networks are not expected to work better, then what is the point? In general we would expect equivariant networks to work better on data with the appropriate invariances, as demonstrated in many previous works. A more likely explanation for why the network is not working well yet is that it has become a bit too complicated, using too many components that deviate far from the existing engineering best practices (i.e. CNNs with relus, batchnorm and skip connections, etc.). For comparison with related work, results in MNIST-rot would be nice to see. This dataset has only 12k training samples. The reported number of 98.42% (obtained from 60k training samples, if I understand correctly) is not really competitive here (SOTA is 0.7% error [5]). Results on non-MNIST datasets would also be very welcome. [1] Hinton et al. 2011, Transforming Auto-encoders [2] Sabour et al. 2017, Dynamic Routing Between Capsules [3] Hinton et al. 2018, Matrix Capsules with EM Routing [4] Cohen et al. 2016, Steerable CNNs [5] Weiler et al., 2017, Learning Steerable Filters for Rotation Equivariant CNNs -------------------- Post author response edit: Thank you for the clear headed author response. Although there is a lot to like about this paper, I will maintain my score because I think that the current evidence does not support the effectiveness of the method. I still do not really understand the point about "pose normalization". I look forward to a future version of the paper where this is clarified. Regarding representations of G, I should have been clear that the concept I had in mind is that of a "linear group representation" (https://en.wikipedia.org/wiki/Group_representation), not a parameterization of G. Indeed, the fact that a Lie group G is a manifold (whose points we cannot easily add / average) was the motivation for mentioning group representations. If instead of viewing poses as elements g in G, you view them as vectors v in V (where V is the vector space of a group representation), then one can average pose vectors v without any difficulty. Several papers in the equivariant networks literature now take this approach.

Reviewer 3



Capsule networks are built to preserve variations in the data (equivariance), but they are not guaranteed to necessarily achieve this goal. This paper uses the concepts from the group convolution works to force the pose vectors and the activations to have equivariance and invariance, respectively. They establish several results guaranteeing such properties for the proposed variation of the capsule networks. A toy experiment is conducted to confirm the findings. Strengths: 1. Formalization of invariance and equivariance in capsule networks. The paper explicitly enforces the invariance for the activations and equivariance for the poses. I consider this part valuable, because it is written in clear mathematical language, compared to the original capsule network papers. 2. Interpreting the resulting group capsule network as a sparse tree representation of group equivariant network. This part argues why the idea of exactly equivariant capsule network might be useful. The authors answer is that it provides a sparse tree representation of a counterpart equivariant network. This result is also insightful. Weaknesses: 1. The key question is “Do we need to make capsule networks explicitly equivariant?” My answer is “most likely no”. Because the idea of capsule networks is not to push for a specific and restricted type of equivariance. It allows freedom in the pose vector and choses the routing algorithm such that the data speak for itself and the pose matrices automatically capture the “right” variances in the lower level features. 2. Continuing the previous comment, aren’t we unnecessarily restricting the capsule networks? Unfortunately, in the experiments the group CapNet is deeper than the simple CapNet and the authors do not provide the number of parameters in the group CapNet. So, we cannot judge how much enforcing the equivariance strictly is beneficial. Because the original CapNet does capture some variations in the data. 3. Unfortunately, the authors omit a key baseline in the experiments: the group-CNN. Not only it is not clear how much extra accuracy is due to CapNets, but also, they do not validate their argument in Section 4. I certainly enjoyed reading this paper, mainly because of the insights and rigor that it brings to the capsule networks idea. Unfortunately, there are doubts about about necessity of this idea and the experiments do not provide any solid evidence either. ======================== After reading the authors' rebuttal: I still do not understand why the proposed model is not directly comparable to the original G-CNN. In the rebuttal, the authors talk about keeping interpretability. But the current version of the paper does not have any experimental validation of these claims. As I mentioned in the original review, I like this paper, but I think it is not ready for publication at the current shape.